# Exploring the Effect of Different Storage Conditions on the Aroma Profile of Bread by Using Arrow-SPME GC-MS and Chemometrics

**DOI:** 10.3390/molecules28083587

**Published:** 2023-04-20

**Authors:** Samuele Pellacani, Marina Cocchi, Caterina Durante, Lorenzo Strani

**Affiliations:** Department of Chemical and Geological Sciences, University of Modena and Reggio Emilia, 41125 Modena, Italy

**Keywords:** aroma, bread, SPME-Arrow, chemometrics

## Abstract

In the present feasibility study, SPME Arrow-GC-MS method coupled with chemometric techniques, was used for investigating the impact of two different storage conditions, namely freezing and refrigeration, on volatile organic compounds (VOCs) of different commercial breads. The SPME Arrow technology was used as it is a novel extraction technique, able to address issues arising with traditional SPME fibers. Furthermore, the raw chromatographic signals were analysed by means of a PARAFAC2-based deconvolution and identification system (PARADISe approach). The use of PARADISe approach allowed for an efficient and rapid putative identification of 38 volatile organic compounds, including alcohols, esters, carboxylic acids, ketones, and aldehydes. Additionally, Principal Component Analysis, applied on the areas of the resolved compounds, was used to investigate the effects of storage conditions on the aroma profile of bread. The results revealed that the VOC profile of fresh bread is more similar to the one of bread stored in the fridge. Furthermore, there was a clear loss of aroma intensity in frozen samples, which could be explained by phenomena related to different starch retrogradation that occurs during freezing and refrigeration. However, considering the limited number of investigated samples, this study must be considered as a proof of concept; a more statistically representative sampling and further examinations of other properties, such as bread texture, need to be performed to better understand whether samples destined for eventual analysis should be frozen or refrigerated.

## 1. Introduction

Bread is a staple food consumed globally, and it is an important component of the Mediterranean diet [1]. Bread is a source of nutrients, including proteins, carbohydrates in the form of starch, dietary fiber, vitamins, and minerals [2]. It also contains antioxidants, such as phenolic compounds, that play a role in protecting the body against oxidative stress [3]. The assessment of bread quality is a complex process that involves several sensory parameters, such as colour, taste, smell, volume, and texture. Maintaining consistent quality and flavour is crucial in the industrial production of bread, particularly when different countries, plants, and raw materials are involved. Among the different sensorial parameters, previous studies have shown how the aroma of bread is an important factor that influences consumers’ choices when purchasing bread [4].

The analysis of bread aroma is an area of research in constant evolution, and many techniques and different approaches are available to chemically characterize the volatile organic compounds (VOCs) of bread [5]. Chemical characterisation of VOCs is generally carried out by means of headspace analysis protocols, which optimize VOCs sampling/extraction conditions, produce a signal typical of the investigated sample, identify the various chemical compounds, and eventually quantify the analytes using appropriate standards. Several methods have been developed to extract, concentrate, and sample food flavour patterns, such as essential oil extraction with solvents, collection of released volatile molecules by direct headspace collection, Solid-Phase Micro Extraction (SPME), and Head-Space Sorptive Extraction (HSSE) [6,7,8]. Most of these methods employ gas chromatography (GC) to separate the collected molecules and mass spectrometry (MS) to identify the chemical species. Over 300 analytes belonging to different classes of chemical compounds have been identified in the aroma profile of bread, including carboxylic acids, aldehydes, ketones, alcohols, esters, etc. [9]. The presence and intensity of these analytes are mainly due to the raw materials used and the bread production process, which involves three essential steps: Mixing of ingredients and preparation of dough, fermentation of dough, and baking. Enzymatic reactions that occur during dough fermentation and lipid oxidation reactions of flours influence the chemical profile of bread crumbs, while thermal reactions during the baking process, such as Maillard reactions and caramelisation of sugars influence the chemical composition of bread crust [10].

However, due to practical limitations, freezing wheat bread samples is a common practice before analysis to preserve volatile compounds [11]. Despite this widespread practice, there is limited research on the impact of freezing on the volatile profile of the bread. This is particularly important considering the ongoing challenge in the baking industry to extend the shelf life of bread, as short shelf life continues to result in significant economic losses.

A critical aspect when investigating the influence of storage conditions on bread VOCs can be associated to the methodology used for volatile compounds sampling [11,12]. This should ensure an efficient VOCs extraction in order to highlight any possible variation during the storage conditions. Indeed, previous studies have shown how VOCs may undergo different changes during storage at room temperature or in the freezer [11]. In particular, in both cases a loss of volatile compounds compared to fresh bread was observed. However, freezer storage seems to maintain a better aroma quality of bread compared to room temperature storage (up to 1 week), despite losses higher than 20% were determined for volatile compounds related to lipid oxidation (hexanal, 1-hexanol, 1-octen-3-ol, etc.). Inferior losses (lower than 20%) were observed for alcohols arising from fermentations (2,3-methyl-1-butanol, phenylethyl, and benzyl alcohol).

Different extraction methodologies can be carried out according to the physicochemical characteristics of the VOCs under investigation [11,12]. Pico et al. used two different extraction methodologies in their studies [11]: Solvent extraction for monitoring almost all volatiles and static headspace for ethyl acetate and ethyl alcohol. In this study, we tested the possibility of using a novel device tool for sampling bread VOCs, i.e., SPME-Arrow, which is a new extraction technology recently employed in the analysis of volatiles in food materials [13,14,15,16]. SPME Arrow was developed to address issues that arise with traditional SPME fibers, including limited mechanical strength, poor reproducibility, and small extraction phase volumes [6]. Previous scientific studies have shown that the use of SPME Arrow fiber results in higher response rates compared to traditional fibers [15,17]. Furthermore, SPME-Arrow performance in terms of sensitivity and reproducibility was tested in the chemical characterisation of VOCs of different bread samples [18]. This could be also attributed to the larger phase volume of SPME Arrow fiber, which enables a greater volume/mass of analyte to be collected.

Therefore, in this paper, the impact of two storage methods (freezing at −20 °C and cooling at 4 °C) on the bread aroma was investigated by using the novel SPME Arrow-GC-MS method to concentrate and extract the VOCs. Moreover, an untargeted approach, based on PARAFAC2 deconvolution and identification system (PARADISe approach) [19,20,21,22], was used to analyse the raw chromatographic data, which can be highly complex. Once the resolved peak area’s values were obtained through PARADISe, the effects of storage conditions were evaluated by applying Principal Component Analysis.

PARADISe has been shown to be a powerful tool for the simultaneous analysis of chromatographic datasets and has been very recently profitably applied to unravel VOCs profile in bread samples [18]. PARADISe, with respect to PARAFAC2, has been developed to automatise the process of GC-MS data array decomposition and pure profile resolution [19,20,21,22]. In fact, PARADISe guides users to define suitable intervals in the chromatograms, and then manages all the required steps from data visualisation to creating a comprehensive list of identified compounds for the entire set of chromatograms.

Although previous studies have looked at the flavour pattern and concentrations of aromatic compounds in frozen bread [11,12], to our knowledge, there are no comparative works that also consider refrigeration. However, considering the limited number of samples, in terms of number of replicates of fresh samples as well, this study must be considered as a proof of concept. More sound and statistically relevant sampling must be performed in order to gain a deep knowledge on which storage method could be able to best exhibit the volatiles of fresh bread together with further examination of other properties related to chemical and physical features of breads.

## 2. Results and Discussion

### 2.1. Peaks Deconvolution and Resolution

In this study, the volatilome of three commercial bread samples (Section 3.1) was analysed using SPME Arrow-GC-MS. These samples were analysed to study the temporal trend of the aromatic pattern as a function of the two different preservation methods: Refrigerator and freezer. The aromatic profile of the three samples was chemically characterised as soon as the bread package was opened and, after 1, 3, and 4 weeks of storage of the sample both in the refrigerator and in the freezer.

Headspace profiles obtained for all samples are reported in Appendix A, while Figure 1 depicts the final signals used in the further data analysis, where the first 7 min and the last 23 min of the chromatograms were excluded, as they only represented the peaks due to adsorbed atmospheric gases and the absence of relevant analytes, respectively. 

The VOCs chromatographic signal, for each bread sample, was arranged as a three-dimensional array having the following three dimensions: Elution profiles (first mode), the recorded mass fragment (i.e., mass spectra as second mode), and bread samples (third mode). The chromatograms were aligned using a routine implemented in PARADISe [21]. The chromatograms were processed in PARADISe, resulting in the identification of 38 analytes (with a Match Factor > 85 and after excluding non-significant compounds [23]) from 78 intervals selected as explained in Section 3.3. The validity of the PARAFAC2 models was then validated by examining the raw data, estimated elution profiles, and mass spectra, as well as the residuals for random distribution and unmodeled peaks.

The workflow in PARADISe starts by determining the right number of components to include in the PARAFAC2 modelling. In this study, the number of explored factors ranged from 1 to 7, and the maximum number of iterations to reach convergence was set as equal to 3000 (default parameter). The first used criterium was to select the smallest number of components that resolve the highest number of peaks [21] and, at the same time, describe the maximum variance (Fit %). To achieve this, PARADISe utilises a machine learning routine that can identify chromatographic peaks. However, the final decision of the number of components to be used for peak integration and for generating the report is due to the user. For the sake of clarity, in Figure 2, an example of the graphical interface is reported, which summarised the information used for modelling one of the selected intervals (retention time, Rt from 24.87 to 25.06 min) considered in this study. The box in Figure 2a reports the explained variance (Fit%) for each PARAFAC2 model as a function of the number of components. The intensity of the grey colour (Figure 2b) is due to the number of peaks resolved by the respective components. The TIC signals of the selected interval and the resolved components by the model are reported in Figure 2c,d, respectively. From the box of Figure 2d, the user can evaluate and choose the components that effectively represent the chromatographic peaks. In the reported example, four components are selected to build the model since they are the best compromise in terms of explained variance and number of resolved compounds (Figure 2a,d). From the investigation of Figure 2d, three of these components correspond to chromatographic peaks (purple, yellow, and red signals) and one (blue signal) represents the baseline. Finally, the peaks are putatively identified, comparing the resolved mass spectra with the spectra in the available libraries (NIST 08 and Wiley 275), to be 2-pentyl-furan, ethyl hexanoate, and octanal, respectively.

This procedure was applied iteratively to each of the 78 selected intervals, and the identified analytes were refined by eliminating a series of aliphatic and aromatic hydrocarbons that, according to the literature [23], do not significantly contribute to the aroma. This resulted in the selection of 38 final compounds (Table 1) identified by NIST spectra. 

### 2.2. Evaluation of Volatile Compounds during Storage

The areas obtained for the 38 volatile compounds individuated in fresh, frozen, and refrigerated samples are provided in Appendix A. For the sake of clarity, Figure 3 depicts the evolution of the main groups of volatile compounds in B-bread samples (fresh, refrigerated, and frozen samples) as a function of time (1, 3, and 4 weeks) and storage methods. The results are the sum of the peak areas of the ketones (Figure 3a), acids (Figure 3b), esters (Figure 3c), alcohols (Figure 3d), and aldehydes (Figure 3e).

As shown in Figure 3, there are different trends for the various analytes depending on the storage methods.

In particular, regarding freezing modality (dashed lines), the total areas of acids, esters, and alcohols followed a general decreasing trend during time. Ketones and aldehydes increase after 1 and 3 weeks, respectively, and then decrease in the 4th week. As far as the decrease in aldehydes after 1 week is concerned, it can be justified by their reduction to alcohols or oxidation to acids. On the other hand, their increase after 3 weeks, could be explained by lipid oxidation, probably favored due to the presence of olive oil in the recipe.

As far as refrigeration storage is concerned (solid line), the key differences between the evolutions of the volatile compounds are achieved during the 3rd week, where it can be possible to note an increase in all the areas of the investigated compounds. In particular, acids, ketones, and alcohols present higher or at least equal values in comparison with the initial values of the respective fresh samples. These trends could be also explained through the oxidation or reduction reactions of aldehydes probably favored during the refrigeration storage.

In any case, for both storage modalities, the total areas of the main species after 4 weeks undergo a decrease compared with the fresh product. As for the total areas of esters, alcohols, and ketones, this decrease is less pronounced in samples stored in fridge. Although the variation in some cases could not be significantly pronounced, some losses may reduce the perceptions of some species that have low odor threshold values in water (OTV), such as ethyl hexanoate (OTV in water [24]: 1 μg kg^−1^), 2-heptanal (OTV in water [25]: 0.06 μg kg^−1^), octanal (OTV in water [24]: 0.7 μg kg^−1^), 3-methyl-butanal (OTV in water [24]: 0.2 μg kg^−1^), etc.

In order to achieve a simultaneous information on the evolution of the individuated volatile compounds in monitored bread samples, all the obtained areas were organised in a two-dimensional matrix (21 bread samples × 38 areas), autoscaled and analysed by PCA analysis (model built with two PCs, explaining 59% of total variance).

The analysis of the scores of PC1 vs. PC2 (Figure 4) shows a trend in the distribution of scores for all three types of bread (A, B, and C). Indeed, considering each type of sample, bread stored in a refrigerator has higher scores in PC1 compared to those stored in a freezer. This trend is visible as a black arrow in the graph, and it goes from positive to negative values as the samples move from being stored in a refrigerator at 4 °C to being stored in a freezer at −20 °C.

The distribution of PC1 scores for type A bread samples indicates a higher degree of variation in the aromatic compounds present in these samples across the two storage conditions. On the other hand, the range of score values for the other two types of bread samples, which are mainly differentiated on PC2, are comparatively narrower, suggesting a lower degree of variability in their aromatic profile. This observation could be attributed to differences in the composition of the bread samples’ function of the type of flour used, the baking conditions, and the fermentation process. In addition, the analysis shows that fresh samples (A_0, B_0, and C_0) exhibit a more similar aromatic profile to those stored at 4 °C suggesting that this storage condition could help in better preserving the fresh aroma of bread for a longer period compared to under freezing storage. A particular behaviour could be noticed for samples A, B, and C stored in the freezer for 3 weeks, that present a noticeable decrease for both PC1 and PC2 score values with respect to the other samples belonging to the same type of bread.

From the PC1 vs. PC2 loadings plot in Figure 5, it can be observed that A samples, stored in the fridge and located on the positive side of PC1 scores, are characterised by a higher content of some chemical compounds, namely benzaldehyde, 2-heptanone, 3-hexen-1-ol acetate, 2-methyl-1-butanol, acetic acid ethyl ester, 2-methyl-1-propanol, butanoic acid, and 3-methyl-ethyl ester. On the other hand, B and C samples, stored in the fridge, are differentiated from other bread samples by compounds, such as octanoic acid ethyl ester, 2-pentyl-furan, benzeneethanol, ethyl hexanoate, propanoic acid 2-hydroxy-ethyl ester, 3-methyl-butanal, acetic acid, and hexanoic acid.

However, from a deeper investigation of Figure 5, it is possible to note a clear trend, indicated by the arrow, which highlights a general decrease in the concentration of some VOCs from samples stored in the refrigerator or just opened, to those stored in the freezer. A plausible explanation for these differences could be found in phenomena influenced by the retrodegradation of starch. Starch is a complex natural polymer composed of amylopectin and amylose [26]. Retrogradation of starch is a temperature- and time-dependent phenomenon involving the rearrangement of starch molecules into an ordered partially crystalline structure [11]. The recrystallisation kinetics of the two starch polymers differ markedly [12]. Amylose recrystallises significantly faster than amylopectin, as a result, most stale models see changes in amylopectin as the main cause of firming crumbs [27]. The relationship between retrodegradation and the production of volatile compounds has not yet been fully investigated. Despite this, it has been hypothesised that the interactions between the hydroxyl groups of the amylose and amylopectin chains due to retrogradation minimize the interactions with volatile compounds [11]. It has also been demonstrated that freezing temperatures, such as −20 °C are able to accelerate the recrystallisation process. Therefore, it can be assumed that, given the greater retrodegradation and the lower interactions between starch and volatile compounds, the marked decrease in volatile compounds is attributable to their loss during the freezing step. In the case of samples stored at 4 °C, it is evident that the temperature fails to significantly accelerate the retrodegradation, leading to better maintenance of the aroma. Finally, freezer-stored samples show higher concentrations of some long-chain aldehydes (octanal, nonanal, 2-nonenal, decanal) that could be derived from the oxidative degradation of lipids [24,28].

## 3. Materials and Methods

### 3.1. Samples and Storage Conditions

In this study, three types of sliced bread (A, B, and C) were purchased from a large retail store and analysed to assess their aroma profiles during storage at different temperatures. The labels on each package of bread provided detailed information about the ingredients and preparation methods used to produce each sample. Sample A was made with re-milled durum wheat semolina (67.1%), water, extra virgin olive oil (2.8%), yeast, salt, sugar, wheat gluten, malted barley flour, and type “0” soft wheat flour. Sample B, on the other hand, was made with type “0” soft wheat flour, water, extra virgin olive oil (2.3%), dextrose, brewer’s yeast, salt, and malted barley flour. Sample C contained re-milled durum wheat semolina (60%), water, mother yeast (15%, made with type “0” soft wheat flour and water), extra virgin olive oil (2.9%), brewer’s yeast, salt, sugar, and flour of malted barley. They were all surface treated with ethyl alcohol. The bread samples were composed of seven slices, with one slice analysed immediately (day 0), three slices frozen at −20 °C, and three slices stored at 4 °C. After 1, 3, and 4 weeks of storage, a slice of each type of bread was brought back to room temperature. To prepare the samples for analysis, approximately 1 cm of crust was removed from each slice. The remaining crumb was minced by hand with a cutter to obtain homogeneous pieces to the greatest extent possible, and a ground sample of around 1.0 g (the masses are listed in Appendix A) was transferred into a 20 mL glass vial with an aluminium-crimp top closure and blue-silicone/PTFE septum (Chromacol, VWR International Srl, Milan, Italy).

### 3.2. SPME-Arrow-GC-MS

SPME Arrow headspace analysis was carried out with a Divinylbenzene /Carboxen/Polydimethylsiloxane (DVB/CAR/PDMS) (120 µm × 20 mm) fiber with a diameter of 1.1 mm (Restek Corporation, Bellefonte, PA, USA). 

The fiber was attached to an SPME fiber holder (Supelco, Bellefonte, PA, USA) for the extraction procedure. Prior to the experimental analysis, the fiber was preconditioned in the injector port of the GC system based on the recommendations of the manufacturers. The weighted sample (around 1 g, Appendix A) was incubated at 50 °C for 10 min, and then SPME fiber was exposed for 30 min. Following the extraction phase, the fiber was manually transferred to the split/splitless injector of the GC, Agilent Technologies (Santa Clara, CA, USA) 6890 N Network gas chromatograph. The VOCs extraction and sampling procedure was previously optimised by means of an Experimental Design approach [18]. Briefly, the desorption step was performed in splitless mode (3 min) with a split flow of 25 mL/min and by setting the injector temperature at 260 °C. The fiber was desorbed for 13 min. The chromatographic separation was performed with Rxi-1ms capillary column (54 m × 0.25 ID, 1.0 µm) by Restek Corporation with helium as carrier gas at a constant flow rate of 1 mL/min. The GC oven temperature was programmed at 40 °C for 1 min, ramped 4 °C/min to 150 °C, then at 8 °C/min to 250 °C held for 19 min. The GC was interfaced with an Agilent 5973 mass spectrometer. The detection was performed under electron impact (EI) ionisation at 70 eV by operating in the full-scan acquisition mode with an *m/z* scanning range from 25 to 300. The transfer line was heated to 270 °C.

### 3.3. Data Analysis

Given the complexity of the data array, the PARADISe method [18,19] was applied to obtain an efficient and rapid extraction of useful information, i.e., areas of deconvoluted peaks and their putative identification. The chromatograms were analysed by exploiting the capabilities of PARAFAC2 (PARAllel FACtor analysis2) [29]. PARAFAC2 provides an efficient method for simultaneously deconvoluting chromatographic peaks that are co-eluted, have shifted retention times, or have low signal-to-noise ratios for all samples being studied. PARAFAC2 quantifies compounds using the complete pure spectrum and relevant retention time region of each peak. To perform this analysis, we used PARADISe [19,21], a user-friendly software platform that employs PARAFAC2 analysis. PARADISe comes with a graphical user interface (GUI) and all necessary tools for GC-MS data processing, including: (1) Data visualisation, (2) time-based data division, (3) PARAFAC2-based peak deconvolution, (4) validation and extraction of deconvoluted peaks, (5) compound identification using the NIST search engine and NIST mass spectra library or any other library in NIST format, and (6) the generation of a comprehensive metabolite table, including the area of every resolved peak. In this study, the defined intervals were 78, selected in order to enclose, when possible, only a single peak. Finally, NIST 08 and Wiley 275 were quested as libraries. The output matrix of PARADISe was analysed through Principal Component Analysis (PCA).

### 3.4. Software

PARADISe approach was performed by PARADISe software version v.5.98. (http://www.models.life.ku.dk/paradise, accessed on 1 March 2023). PCA was carried out by using PLS_Toolbox 8.9.2 software (Eigenvector Research Inc., Manson, WA, USA) for MATLAB^®^. 

## 4. Conclusions

The aim of this study was to develop an analytical procedure that would provide insights into the effects of frozen and refrigerated storage time on the volatile profile of wheat bread. To achieve this goal, the novel Arrow-SPME fiber coupled with GC-MS was used for the analysis of three different commercial bread samples, stored under both storage conditions for 1, 2, and 4 weeks. The resulting signals were then analysed using the PARADISe approach, which allowed for the attainment of an efficient and rapid extraction of useful information (namely, areas of deconvoluted peaks and their putative identification), followed by Principal Component Analysis.

The obtained results have demonstrated that the aroma profile of the frozen bread was characterised by a reduction in the concentration of several volatile organic compounds with respect to fresh samples and the ones stored in the refrigerator, across all three types of bread that were investigated. However, the samples that were stored in the freezer exhibited higher concentrations of certain long-chain aldehydes, such as octanal, nonanal, 2-nonenal, and decanal. These compounds may have resulted from the oxidative degradation of lipids during the freezing process. Overall, these findings suggest that storing bread in a refrigerator may be a more effective way to preserve the aroma quality of bread over time.

These results could have a significant implication for the baking industry and aroma research, as they shed light on the impact of preservation techniques on the organoleptic properties of bread. This study can also contribute to the development of strategies for increasing the shelf-life of bread while preserving its aroma quality, which is critical to consumer acceptance and purchase decisions. Although further examination of other properties is needed for an in-depth analysis of bread shelf-life and for maintaining quality for other important aspects, such as texture, this study represents an excellent starting point for investigating the impact of time and type of storage on bread aroma.

## Figures and Tables

**Figure 1 molecules-28-03587-f001:**
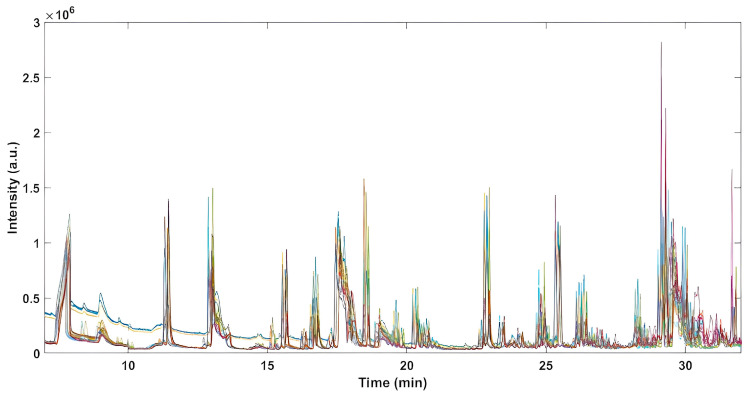
Investigated interval of raw chromatograms (from 7 to 37 min).

**Figure 2 molecules-28-03587-f002:**
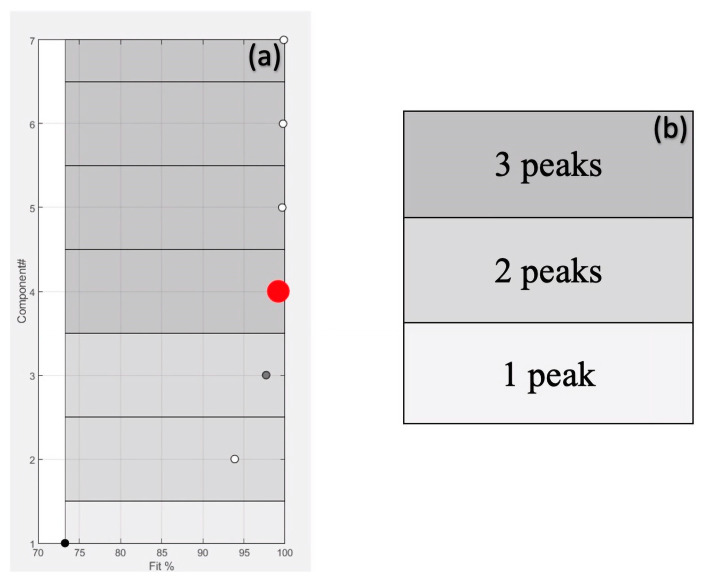
The number of components (y-axis) chosen is depicted by a red circle (**a**). The value of the x-axis indicates the Fit% of the model. The greyscale (**b**) gives immediate information on the number of peaks described by te model. Thanks to the user-friendly GUI, it is possible to look simultaneously at: Total Ion Current (TIC (**c**)), the components resolved by PARAFAC2 (**d**), and the mass spectra of each component (**e**).

**Figure 3 molecules-28-03587-f003:**
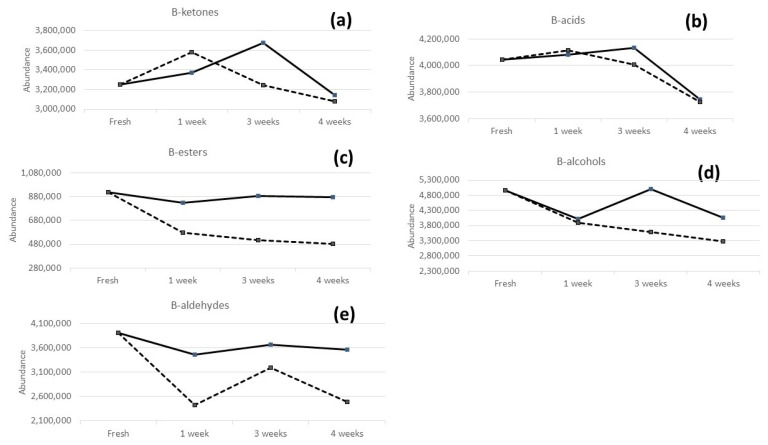
Evolution of the main groups of volatile compounds, namely ketones (**a**), acids (**b**), esters (**c**), alcohols (**d**), and aldehydes (**e**), in B bread samples stored in refrigerator (solid line) and in the freezer (dashed line).

**Figure 4 molecules-28-03587-f004:**
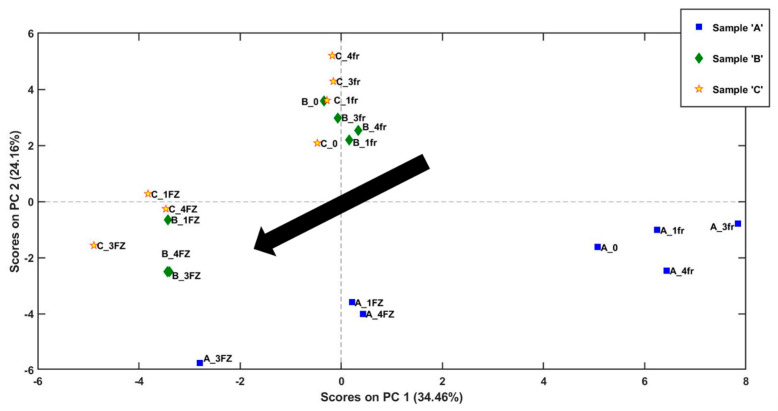
Analysis of VOCs of bread samples sampled with Arrow-SPME. PC1 vs. PC2 scores plot obtained by PCA applied on selected areas from PARADISe analysis. The samples labelled ‘fr’ were stored in fridge, the samples labelled ‘FR’ were stored in freezer. The number on the labels indicates the number of weeks of storage.

**Figure 5 molecules-28-03587-f005:**
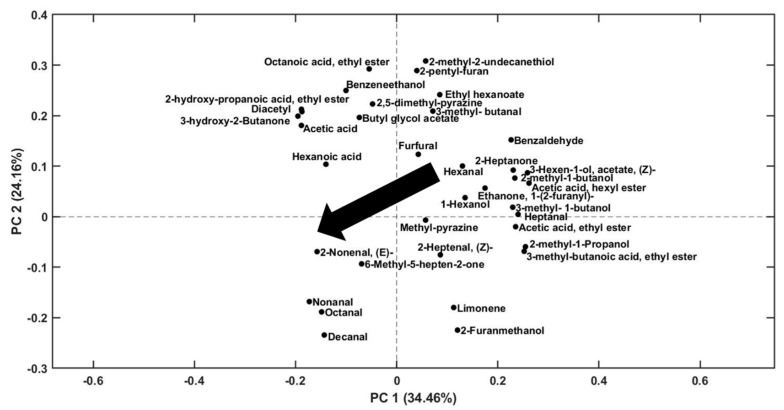
Analysis of VOCs of bread samples sampled with Arrow-SPME. PC1 vs. PC2 loadings plot obtained by PCA applied on selected areas from PARADISe analysis.

**Table 1 molecules-28-03587-t001:** Identified headspace volatile compounds in the bread samples with their retention time and match factor.

Analyte	Retention Time (min)	Match Factor (MF)
Diacetyl	7.53	87.6
Acetic acid	7.87	98.1
Acetic acid, ethyl ester	8.42	89.4
2-methyl-1-propanol	9.01	93.2
3-methyl-butanal	9.70	88.3
2-butanone, 3-hydroxy-	11.40	95.2
3-methyl-1-butanol	12.99	93.3
2-methyl-1-butanol	13.18	95.2
Hexanal	15.64	95
Propanoic acid, 2-hydroxy-, ethyl ester	16.33	93.2
Methyl-pyrazine	16.64	91.1
Furfural	16.76	97.2
2-furanmethanol	18.03	92.3
3-methyl-butanoic acid, ethyl ester	18.34	94.5
1-hexanol	19.08	93.9
2-heptanone	19.82	91.8
Heptanal	20.32	95.7
Ethanone, 1-(2-furanyl)-	20.47	95.8
2,5-dimethyl-pyrazine	20.74	93.6
2-heptenal, (Z)-	22.69	95.5
Benzaldehyde	22.89	97
Hexanoic acid	23.42	93.3
6-methyl-5-hepten-2-one	24.082	95.5
Ethyl hexanoate	24.76	96.9
2-pentyl-furan,	24.85	92.7
Octanal	24.91	89.2
3-hexen-1-ol, acetate, (Z)-	25.03	98.3
Acetic acid, hexyl ester	25.33	92.6
Limonene	26.95	96.1
2-Octenal	27.19	93.5
Butyl glycol acetate	28.29	90.3
2-methyl-2-undecanethiol	29.13	91.2
Nonanal	29.22	88.4
Benzeneethanol	29.62	97.7
2-nonenal, (E)-	31.19	90.1
Octanoic acid, ethyl ester	32.42	87.9
Decanal	32.73	88.8
2(3H)-furanone, dihydro-5-pentyl-	36.46	90.1

## Data Availability

Not applicable.

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
