# Peer review of "Exploring the Effect of Different Storage Conditions on the Aroma Profile of Bread by Using Arrow-SPME GC-MS and Chemometrics"

_molecules, 2023, doi:10.3390/molecules28083587_

Round 1
Reviewer 1 Report
The manuscript # molecules-2283564 deals with the examination od the affect of two different storage conditions, namely freezing and refrigeration, on volatile organic compounds (VOCs) of different commercial breads. The topic is interesting as the authors state that
“However, due to practical limitations, freezing wheat bread samples is a common 56 practice before analysis to preserve volatile compounds. Despite this widespread 57 practice, there is limited research on the impact of freezing on the volatile profile of the 58 bread. This is particularly important considering the ongoing challenge in the baking in- 59 dustry to extend the shelf life of bread, as short shelf life continues to result in significant 60 economic losses.”
The manuscript is also well written and well organized and I would recommend its puplication in Molecules. Some mino revisions are needed before publication as follows;
Line 77-83, the authors should elaborate PARADISe approach little bit in Introduction section.
Fig. 1 is not understandable where is the last 23 minutes of the chromatograms?
Section 3.2 needs ref. and the column information to be used in GC is misssing
Reviewer 2 Report
The abstract focuses more on new technology than findings, whereas the introduction and results are the reverse. Adjust
44-55: Either here or elsewhere describe some of the specific chemical products and both challenges with analysis or expect changes during storage
62-83- Excessive description of study objectives. Should be limited to a paragraph at most with some of the points made relying on what others have and have not done?
77-99: Appropriate in the intro and discussion to make clear that Paradise has been extensively used for this type of work. As written, this gives the impression that this is a novel aspect of this work. If there are novel aspects to what was added to the workflow, make it clear what is commonplace, versus what you have added and the benefit.
223-228- Clarify replications, I am reading this as Ambient n=1, refrigerated n=7, frozen, n=7, and am not seeing any reference to replications. So for each of the three bread samples, a total of 15 runs were conducted.
233-250: How were these parameters decided upon? Previous research with SPME Arrow on solid samples? Or was there an optimization portion of this work?
238- Around one gram? Or 1 gram. Or was there a difference in the weight of different sample preparations?
Results overall: In part this manuscript seems to be a methods paper. As such it should touch on classic method validation aspects such as reproducibility, interference, linearity. Untargeted analysis is a powerful tool, but without being coupled with additional validation it is hard to say that this truly characterized the compounds of importance. If there is supporting work, please include it.
Bread storage: In the abstract, it appears as if this study is aiming to identify whether samples destined for eventual analysis should be frozen or refrigerated to best exhibit the volatiles of fresh bread. However, it appears there was not replication for the fresh bread, making it more of a comparison of refrigerated or frozen. Some additional work needs to be done to finish this comparison, PCAs give great insight into what to focus on, but as a reader, I want to know specifically if I am going to conduct a study on bread, which volatiles are enhanced or degraded with one storage method versus another. As well as what are those relative differences. A small but consistent change may drive a PCA component, but from a practical standpoint be inconsequential.
266-270: Many of the compounds tentatively identified by NIST have spectra very close to related compounds making it hard or not impossible to definitively identify them without additional information such as the retention index, comparison to an authentic standard, or relative elution to known compounds. If identification truly only was done by NIST spectra make this very clear throughout the manuscript. Also as the purpose of this study was to identify differences related to bread quality some attempt at quantification, is warranted. Even if just putting areas as equivalents to a given standard, to understand whether a given compound is anywhere near the sensory threshold. Or merely present.
